# Training Language Models on the Knowledge Graph: Insights on Hallucinations and Their Detectability

**Jiri Hron**[*†]    **Laura Culp**[*]    **Gamaleldin Elsayed**[*]    **Rosanne Liu**[*]

Ben Adlam    Maxwell Bileschi    Bernd Bohnet    JD Co-Reyes    Noah Fiedel
C. Daniel Freeman[◇]    Izzeddin Gur    Kathleen Kenealy    Jaehoon Lee[◇]
Peter J. Liu[◇]    Gaurav Mishra    Igor Mordatch    Azade Nova    Roman Novak[◇]
Aaron Parisi    Jeffrey Pennington    Alex Rizkowsky    Isabelle Simpson
Hanie Sedghi    Jascha Sohl-dickstein[◇]    Kevin Swersky    Sharad Vikram
Tris Warkentin    Lechao Xiao    Kelvin Xu

**Jasper Snoek**[*]    **Simon Kornblith**[*†◇]

Google DeepMind

## Abstract

While many capabilities of *language models* (LMs) improve with increased training budget, the influence of scale on hallucinations is not yet fully understood. Hallucinations come in many forms, and there is no universally accepted definition. We thus focus on studying only those hallucinations where a correct answer appears verbatim in the training set. To fully control the training data content, we construct a *knowledge graph* (KG)-based dataset, and use it to train a set of increasingly large LMs. We find that for a fixed dataset, larger and longer-trained LMs hallucinate less. However, hallucinating on $\leq 5\%$ of the training data requires an order of magnitude larger model, and thus an order of magnitude more compute, than Hoffmann et al. (2022) reported was optimal. Given this costliness, we study how hallucination detectors depend on scale. While we see detector size improves performance on fixed LM's outputs, we find an inverse relationship between the scale of the LM and the detectability of its hallucinations.

## 1  Introduction

Despite rapid progress in generative and predictive capabilities, hallucinations remain a significant challenge for large language models (Gemini, 2023; OpenAI, 2023). Although researchers have carefully studied "scaling laws" (Kaplan et al., 2020; Hoffmann et al., 2022)—an empirical phenomenon where LM performance improves as dataset and model size increase—little is known about how hallucinations depend on scale. To fill this gap, the first issue at hand is to precisely define and quantify hallucinations. However, in natural language setting this is very hard as language expressions can be ambitious, and the exact knowledge content in training data is notoriously unclear.

Knowledge graph (KG), on the other hand, offers full controllablility of its factual content: it is straightforward to query whether a generated fact from an LM indeed exists in the dataset or not, hence offering quantifiable measure of hallucination. Training LMs on KG allows us to study the extent to which LMs misrepresent their training data, and how this phenomenon depends on scale.

---

[*]Core contributors. Correspond to jirihron@google.com
[†]Equal contribution.
[◇]Work done while at Google DeepMind.

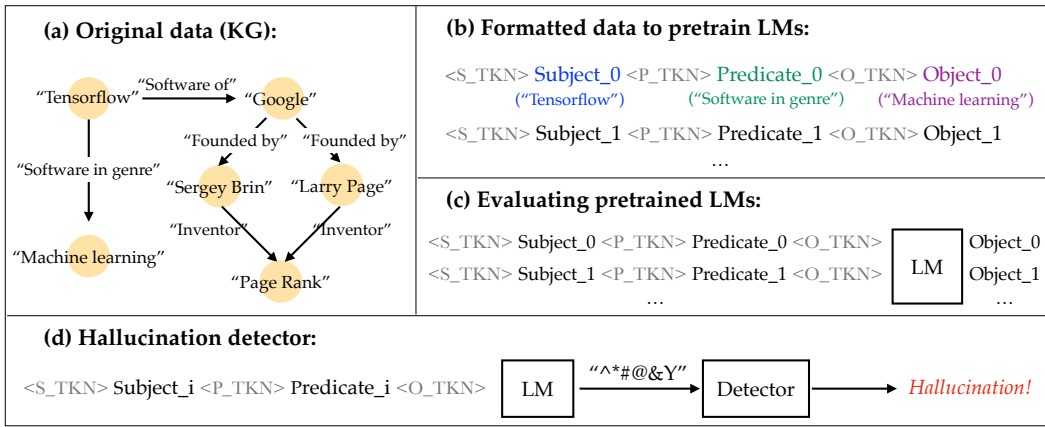

Figure 1: **Data and the training pipeline.** `<S_TKN>`, `<P_TKN>` and `<O_TKN>` are special tokens indicating subject, predicate, and object, respectively. **(a)** The original data exist in the form of a Knowledge Graph (KG), where nodes representing subjects and objects are connected by predicates (arrows). **(b)** The KG is then formatted into triplets: subject, predicate, object, and further prefixed with special tokens indicating their identity. Such formatted data are used to pretrain autoregressive LMs with the common next-token-prediction loss. **(c)** Pretrained LMs are evaluated by prefixing with subject and predicate alongside special tokens to predict objects. **(d)** On top of pretrained LMs, detectors are trained to detect the presence of hallucinations during generation.

We therefore train a number of increasingly large LMs from scratch on data extracted from a KG. A KG contains knowledge about the world in the form of [subject, predicate, object] triplets (Figure 1(a)). We concatenate the triplets into strings, and use these to train the LMs by optimising the auto-regressive log-likelihood (Figure 1(b)). When evaluating, we prompt the model with a subject and predicate, and let it complete the object (Figure 1(c)). A generation is considered a hallucination if the prediction does not match any object that appears with given subject-predicate pair in the training set.

The speciality to this setup is that each unique piece of information is only seen once per epoch: each triplet contains a unique piece of information that is distinctive from any other triplet. This is unlike any traditional LMs trained over natural language, where the same information content may appear multiple times even in deduplicated datasets, due to the flexibility of natural language. This difference engenders three major observations that deviate from our current understanding of LM (pre)training on text.

First, multi-epoch training is necessary, as one might expect in this setting, for the training loss to converge and hallucination rate to go down to an acceptable level. Secondly, the scaling between the model size and the number of training tokens is different from what is currently considered optimal (Hoffmann et al., 2022); LMs several times larger are needed to eliminate training set hallucinations. What's more, counter to standard scaling laws (Kaplan et al., 2020; Hoffmann et al., 2022), the non-duplication of triplets also means that increasing the training set size (adding more pieces of information) leads to worse performance for any fixed LM size and training length. To decrease the training set hallucination rate, one can train for longer and set the generation temperature to zero, but this often comes at the cost of worse generalisation to data not seen during training. Thus, there is a trade-off between training set hallucination rate and other LM capabilities.

Equipped with a range of LMs trained from KG with various hallucination rates, we then investigate methods that prevent hallucinations by detecting—and optionally also correcting—them post-hoc, as well as the scaling behaviour of such methods (Figure 1(d)). While we do find that larger detectors perform better, and that detection accuracy improves with the LM size and length of pretraining, there is a caveat: the improvement in accuracy with LM scale is confounded by the already lower hallucination rate of the more powerful LMs. Examining the detectors' precision-recall (PR) curves, we find an *inverse* relationship between the LM size and the detector's area under the PR curve (AUC-PR). In other words,

| Dataset | Language Model | | Detector | | Size (total triplets) |
|---------|----------------|----------------|----------------|----------------|------------------------|
|         | Trained on?    | Eval-ed on?    | Trained on?    | Eval-ed on?    |                        |
| FVS, 1%   | ✓ | ✓ | ✓ | ✗ | 6,505,590 |
| FVS, 10%  | ✓ | ✓ | ✓ | ✗ | 65,150,414 |
| FVS, 100% | ✓ | ✓ | ✗ | ✗ | 555,969,021 |
| PVS, 10%  | ✓ | ✓ | ✗ | ✓ | 159,502 |
| PVS, 100% | ✓ | ✓ | ✗ | ✓ | 1,606,058 |
| IVS       | ✗ | ✓ | ✗ | ✓ | 19,311 |

Table 1: **Datasets created with different levels of visibility and scale.** The *fully visible set* (FVS) is used for training of both the LMs and detectors (except for the 100% version). The *partially visible set* (PVS) is only used for the training of LMs, not the detectors. The invisible set (IVS) is a fully held-out test used only for some of the evaluations. For each visibility level, different dataset sizes are created by sampling 1% or 10% of the respective full set. The total number of triplets contained in dataset version is shown. We always evaluate LMs on both training and test sets, while only evaluate detectors on held-out test sets.

| Dataset | Training epochs | | | | | |
|---------|---|---|----|----|-----|-----|
|         | 1 | 2 | 10 | 20 | 100 | 200 |
| FVS + PVS, 1%   | ✓ | ✓ | ✓ | ✓ | ✓ | ✓ |
| FVS + PVS, 10%  | ✓ | ✓ | ✓ | ✓ | ✗ | ✗ |
| FVS + PVS, 100% | ✓ | ✓ | ✗ | ✗ | ✗ | ✗ |

Table 2: **Length of training** for different dataset size. We train a maximum of 200 epochs, on the smallest subset, and as the training set grows, we reduce the training length.

the more powerful the LM, the harder it is to detect its hallucinations. This is despite the fact that we have control over what the model is exposed to, which has been thought to be key to making hallucination detectors work well (Schulman, 2023).

The rest of the paper is structured as follows. We describe the dataset and training of LMs in Section 2, investigate the interplay between hallucinations and LM scale in Section 3, and study the relation between LM scale and hallucination detectability in Section 4. The remaining sections are devoted to the discussion of limitations and takeaways.

## 2   Controlling What an LM Knows

A core challenge in studying LM hallucinations is that we typically do not know what information the model was exposed to during training. Without this knowledge, we cannot determine whether the model output is wrong because (a) it has never been trained on a given fact, (b) it was trained on it but did not memorize it, or (c) the model memorized factually incorrect information that did appear in the training set. To avoid these issues, which are further confounded by various pretraining and finetuning strategies used in state-of-the-art LMs, we train LMs from scratch (Section 2.2) on data specifically constructed to give us perfect control over the information a model sees (Section 2.1). This will later enable us to investigate how model and dataset scale affects hallucinations (Section 3), and their detectability (Section 4).

### 2.1   The Knowledge Graph dataset

We propose using a *Knowledge Graph* (KG) as a way of controlling the information a model sees. KGs are structured, factual data, that are often used within organisations to feed knowledge into various applications. We use KG as it provides a repository of information which is self-consistent, and mirrors the structure of information in the real-world; the hope

is that this mirrored structure will ensure that the character of any hallucinations we see is somewhat similar to hallucinations we would see from models trained on data more typical for LM training. The main benefit to using a KG, however, is that we know exactly what a model has seen, and since it is structured data, we can easily query the data to see if its predictions are correct.

The KG we use (Google, 2012) contains semantic triples: a `Subject`, `Predicate`, and `Object`. We further insert special tokens before each of the `Subject`, `Predicate`, and `Object`, as indicated in Figure 1, and use the concatenated strings to train our LMs. The processing removes the ambiguity of natural language, which makes the task both easier and harder for an LM. The task is easier because the samples are now *structured*: LMs no longer need to pick up the intricacies of grammar and distinguish different phrasings, and can instead just focus on learning facts. It is, however, also harder, because there is very little correlation between data samples, unless they share items, and thus very little positive transfer between learning one fact to the other.

In later sections (Sections 3 and 4), we will be training and evaluating LMs and hallucination detectors. We therefore need to carefully design data splits to fully understand the impact of data. We design datasets that reflect three levels of visibility: 1) a *fully visible set* (FVS) that both the LMs and detectors are trained on, 2) a *partially visible set* (PVS) that only the LMs are trained on, and finally 3) an *invisible set* (IVS) that neither the LM or the detector have seen. We then vary the sizes of FVS and PVS to study the effect of scale.

To construct these three sets of triplets, we perform an i.i.d. split at the subject level. Some subject-predicate pairs are associated with multiple objects (e.g., names of tracks on an album). In these cases, we need to ensure that all objects associated with a given subject-predicate belong to the same set, as otherwise we might label correctly deduced object predictions as hallucinations. A similar issue can exist at the subject level, e.g., age can be deduced from date of birth. Several subject-predicate pairs are associated with hundreds of objects. To simplify evaluations, we remove all subject-predicates linked with more than 20 objects. This eliminates extreme long-tails that would be hard to meaningfully analyse. Table 1 reports the resulting dataset size.

## 2.2 Training LMs on the Knowledge Graph

We trained decoder-only transformer LMs (Vaswani et al., 2017) with varying number of non-embedding parameters (3.15M–1.61B), on various sizes of the KG dataset (1%–100%). The parameters are optimized with respect to the autoregressive cross-entropy loss over formatted strings created from triplets with special tokens (Section 2.1).

Where a single triplet does not fill the context window (256 tokens), we used packing (on average, ~20 triplets fit into the context window). For optimization, we used Adam (Kingma & Ba, 2014) with linear learning rate warmup from 0 to our base learning rate (4K steps), followed by cosine decay down to 5% of the base rate. We varied the total number of steps to study the effect of multi-epoch training (see Tables 2 and 4 for details). The base learning rate is set to a constant divided by the square root of the number of LM's non-embedding parameters. The constant was determined by a hyperparameter search over 2.5, 5, and 10 (due to compute limitations this exploration was not done for all models). The exact learning rates we used can be found in Table 3 in Appendix.

## 3 Hallucination Rate and How It Scales

Scaling laws are an empirical phenomenon where the cross-entropy loss of LMs decay as a power law of the model and training set size (Kaplan et al., 2020; Hoffmann et al., 2022). Since cross-entropy is related to the accuracy of model predictions, one can wonder whether hallucination rates follow a similar trend. Figure 2 shows this is not the case. While for a fixed dataset size, larger, longer-trained models tend to hallucinate less, increasing the dataset size yields a higher, rather than a lower, hallucination rate (top left vs. top right); a similar trend is observed for the cross-entropy loss—evaluated on full triplets, not just the object—in Figure 3. This is because of two factors: (1) many triplets in the KG require

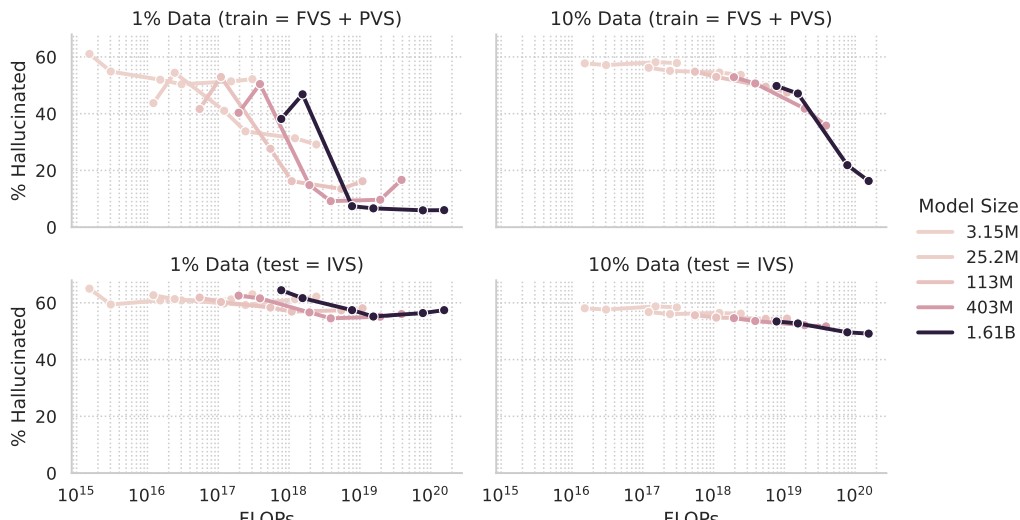

Figure 2: **Hallucination rate per LM training FLOPs** on examples seen (top) and unseen (bottom) during training, on small (left) and large (right) size of data. Each dot is an independent training run with learning rate schedule adjusted to the training length (Section 2.2). Dots correspond to [1, 2, 10, 20, 100, 200] epochs for 1% Data, and [1, 2, 10, 20] epochs for 10% Data. For a fixed dataset, the more FLOPs, the lower the hallucination. In contrast to established scaling laws for loss on text (Kaplan et al., 2020; Hoffmann et al., 2022), performance actually worsens with dataset size (top left vs. top right), as the larger dataset requires learning more facts. Training for 20+ epochs is necessary to minimise hallucinations on seen data (top), but can lead to overfitting to unseen data (bottom), presenting a trade-off between fact recall and ability to generalize. This is even more pronounced at temp = 0.0 (Figure 8). The hallucination rate upticks for 113M and 404M LMs on 1% Data are not mirrored by the training loss (Figure 3), i.e., they are not due to loss divergence.

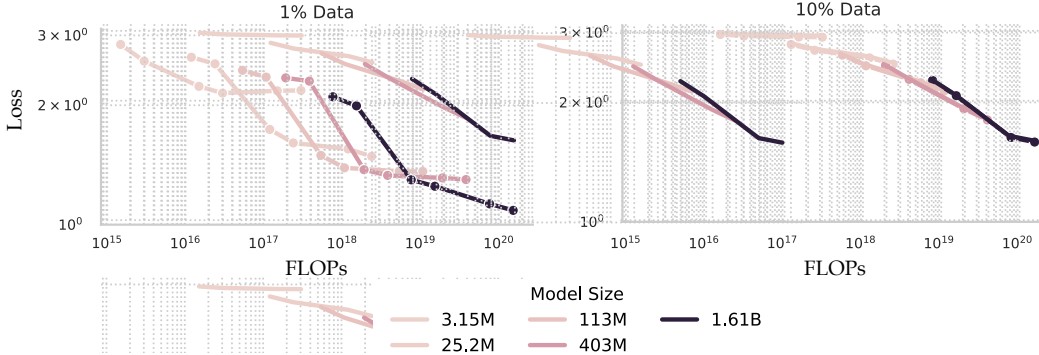

Figure 3: **Training loss per LM training FLOPs.** The y-axis shows per-token average autoregressive cross-entropy loss over *all tokens* (i.e., subject, predicate, and object). Complementing Figure 2, larger models attain smaller loss at a fixed dataset size, but the loss increases when training set size grows.

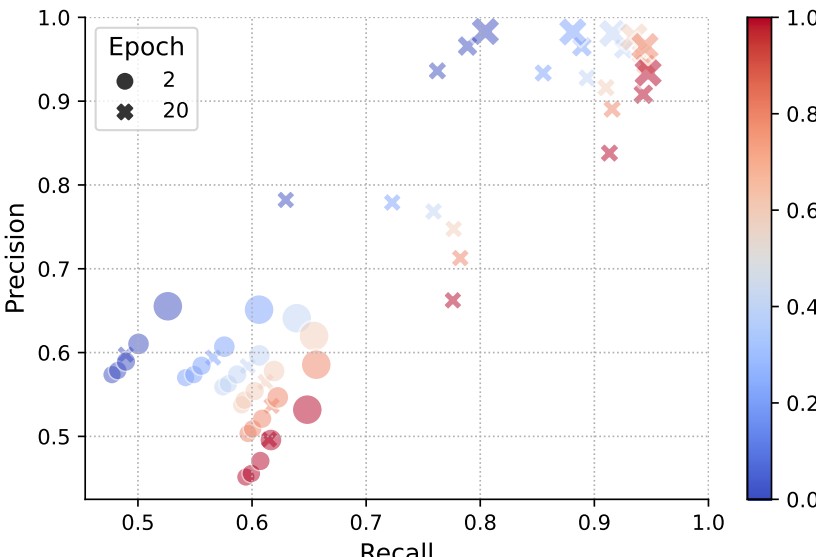

Figure 4: **Precision and recall as function of temperature** on the 1% Data. Marker size represents the number of non-embedding parameters. Marker type the number of epochs for which the LM was trained. For each temperature in [0.0, 0.2, 0.4, 0.6, 0.8, 1.0], we generate 16 object predictions for every subject-predicate, and evaluate precision and recall against the valid completions. The reported numbers are average over all examples. Lower temperatures yield higher precision, higher temperatures yield higher recall.

memorization (i.e., correct answer cannot be inferred from other data points); (2) each triplet appears in the training set only once. The bottom two plots in Figure 2 demonstrate the necessity of memorization; no model attains less than 50% hallucination rate on data not seen at training time. Examples of facts requiring memorization are names of tracks on an album, or dates of birth.

Another consequence of the necessity to memorize and of the lack of repetition is that 20+ epochs are needed to achieve close to minimal hallucination rate for a given LM size. This stands in sharp contrast to the current practice of training LMs for only one or a few epochs (Chowdhery et al., 2023; Hoffmann et al., 2022; Touvron et al., 2023); we show for models trained for fewer (1 or 2) epochs in the beginning of each line in Figure 2.

An unfortunate side-effect of training for 20+ epochs seems to be a decreasing ability to generalise to unseen data (note the eventually upwards slope in the bottom plots). This trend is even more pronounced at temp = 0.0 (Figure 8), presenting a trade-off between hallucination rates and other model performance measures. While even the most capable modern LMs hallucinate (Gemini, 2023; OpenAI, 2023), repetition of facts in their training data likely alleviates the issue, as also suggested in Kandpal et al. (2023). Excessive repetition might however be harmful (Hernandez et al., 2022; Xue et al., 2023; Muennighoff et al., 2023), presenting a challenge for curation of training data.

It is striking that, in our case, even when training on 1% of the full KG, the hallucination rate on *data seen during training* remains ∼5% for the largest and longest-trained LM. This hallucination rate can be pushed down to ∼1.5% by lowering the temperature from 1.0 to 0.0. However, the KG contains a large number of subject-predicate pairs with multiple associated objects (e.g., names of all authors of a particular paper), and reducing sampling temperature leads the model to generate fewer of these objects. In Figure 4, we show how varying temperature affects both *precision*, defined as 1-hallucination rate, and *recall*, defined as the average proportion of objects in the original training data that the model generates at least once when sampling 16 completions for each subject-predicate pair. This highlights an issue with focusing on hallucination rate only: an easy way to hallucinate less is to make fewer claims. A related issue is that finetuning LMs to refuse to answer when uncertain may potentially lead to overly conservative behaviour.

Our best-performing LMs are an order of magnitude larger than the predicted Chinchilla-optimal size for the number of tokens on which we train (Hoffmann et al., 2022), and therefore even larger than the even smaller models geared towards efficient inference (Touvron et al., 2023). This suggests that even if we just wanted LMs to rarely hallucinate on the data that they have seen during pretraining, we would need a computational budget several times higher than what is currently considered optimal. A more efficient alternative may be retrieval-augmentation (Lewis et al., 2020; Borgeaud et al., 2022), or methods for expressing uncertainty or self-correcting (e.g., Dhuliawala et al., 2023; OpenAI, 2023; Li et al., 2023). In Section 4, we study influence of scale on methods that belong to the latter category.

## 4 Hallucination Detectability and How It Scales

### 4.1 Setup

In Section 3, we have seen hallucination rates typically decay with LM size and training length. However, even LMs much larger than currently considered optimal—for given training set size—continue to hallucinate ∼5% of the time on data seen and ∼50% on data unseen during training (Figure 2). Our experiments also exhibit a trade-off between in-distribution and out-of-distribution hallucination rates (Figures 2 and 8). We therefore need to understand whether it is possible to further reduce hallucination rates by other means.

There are many types of alternative interventions, ranging from retrieval to model self-correction (see Appendix A). One promising direction is *hallucination detectors* which try to identify hallucinations either from the LM output itself, or from the LM's internal representations (e.g., Kadavath et al., 2022; Dhuliawala et al., 2023; OpenAI, 2023; Li et al., 2023). Our aim is to understand (i) how the effectiveness of hallucination detectors depends on the scale and training length of the LM they are judging, (ii) what types of detectors perform better, and (iii) if there is evidence other interventions beyond detectors are needed.

We explore two types of detection tasks:

- sentence: The detector ingests both *original* subject-predicate and the *predicted* object tokens, and judges if the object is hallucinated.[1]
- token: The detector takes embedding of a token from a given layer of the trained LM, and is asked to say if it is hallucinated. For a given subject-predicate input, a predicted object token is labelled as hallucinated if it and the tokens preceding it do not match the token prefix of any entry in the KG. The goal is to predict the first hallucinated token; any tokens that come after are discarded from the detector's training and evaluation data.

For each of the tasks, we experiment with two types of hallucination detectors:

- head: Adds a new readout head on top of the pretrained LM that produced the output (remaining LM weights are frozen). For the sentence task, this is equivalent to adding two new tokens to the dictionary (hallucination & non-hallucination), and finetuning the LM readout layer to predict whether the preceding triplet is hallucinated. The same holds for the token task, but the point of prediction changes; we also experiment with training detectors based on token representations from after each transformer block (Figure 10).
- full: Takes the pretrained LM as a starting point, and finetunes all its weights. For this setup, we only consider the case where the readout head is attached to the top layer embeddings in the token task.

---

[1]In preliminary experiments, we tested three different setups: (a) predict 'ahead-of-time', i.e., straight after the subject-predicate (related to Kadavath et al., 2022); (b) predict straight after the last object token; and (c) predict after the end-of-sentence (EOS) special token. While (a) is more efficient in terms of inference costs, it lags behind the other two in terms of hallucination detection accuracy. From (b) and (c), the latter was more stable to train, presumably because it clearly signals to the detector that the LM output is complete. Since our main interest lies in accurate detection of hallucinations, we used setup (c) in all the presented experiments.

The combination of `token` task and `head` detector applied to the top layer embeddings is similar to the approach taken in Kadavath et al. (2022). For `token` and `full`, the results should provide an upper bound on the performance achievable by finetuning the whole LM to say 'I don't know' (IDK), as we do not force the detector to preserve any of the other LM capabilities.[2] For the `sentence` task, the setup is related to techniques based on post-hoc verification, including self-critique (e.g., OpenAI, 2023). The combination with `full` detectors in particular should provide an upper bound on performance achievable by self-critique, as we force the preserve any of the other LM capabilities.[3]

We trained a distinct detector (of each type) for every combination of a pretrained LM and detection task. Training and evaluation data are obtained by generating 5 object predictions for every subject-predicate in the LM training set. This data is split into a detector training (90%), validation (5%), and test sets (5%). The validation set is used for hyperparameter tuning and early stopping, the test set for measuring performance.

For the `token` experiments, we use the Adam optimizer (Kingma & Ba, 2014), with 1K warm-up steps and cosine decay (5K steps for `head`, 20K steps for `full`). Peak learning rates 1e-4 was used for `head`, and 5e-5 for `full` detectors. Training length and learning rates were determined using the validation set, optimizing for high AUC-PR. We did not finetune every possible combination of LM, detector task, and algorithm due to the large size of the Cartesian product (over 300). Visually, the above choices were optimal (or close) for most setups, but may not be optimal.

For the `sentence` experiments, the Adam optimizer alone underperformed for the `full` detectors, so we switched to *linear probing then full finetuning* (LPFT; Kumar et al., 2022). LPFT works in two stages: (i) only the readout layer is optimized, with all other weights frozen; (ii) both readout and other layer weights are optimized. We used the Adafactor optimizer (Shazeer & Stern, 2018) for both stages. In the first, we used a cosine decay schedule with peak learning rate 1e-2 for 10K steps. In the second, we used linear warmup (10K steps) combined with cosine decay (250K steps) with peak learning rate 1e-3. For `head` detectors, only the first stage was used. The number of training steps and learning rates were determined using the validation set, aiming for high AUC-PR. Unlike in the `token` setup, we found it harder to find a common set of hyperparameters that would work for all detectors.

## 4.2 Results

Figure 5 shows how the pretrained LM size affects overall accuracy across tasks and approaches (Section 4.1). As expected, the `full` detectors outperform `head`, as they tend to be more flexible. Since this is also the case for other metrics, we focus on `full` detectors in the rest of this section. The `token` task formulation generally yielded better detector accuracy than the `sentence` task. Detector accuracy also tends to grow with the size of the underlying LM. However, these results are confounded by sensitivity of the accuracy metric to the underlying LM hallucination rate (see Figure 2). For example, if the rate is only 5%, even a naive detector which catches all hallucinations attains 95% accuracy.

We use AUC-PR to assess how well our detectors identify hallucinations. In Figure 6, we observe that: (a) `sentence` task formulation is superior in terms of AUC-PR, and (b) the lower the LM's hallucination rate, the worse the detector's AUC-PR. Per Figure 2, the lowest hallucination rates are achieved by the largest longest trained models. Thus, for a fixed training set size, there is an *inverse relationship* between the FLOPs spend on the LM pretraining and the detectability of its hallucinations.

Figure 7 emphasizes the inverse relationship between LM scale and hallucination detectability, showing the PR curves corresponding to the `sentence` task AUC-PR values in Figure 6. Note the general ordering of the curves, with those corresponding to detectors for the largest LMs being the lowest, and the ones for the smallest LMs being the highest.

---

[2]Upper bound does not hold if there is a synergy between the original LM capabilities and learning to say IDK.

[3]Again, this does not hold if there is a synergy.

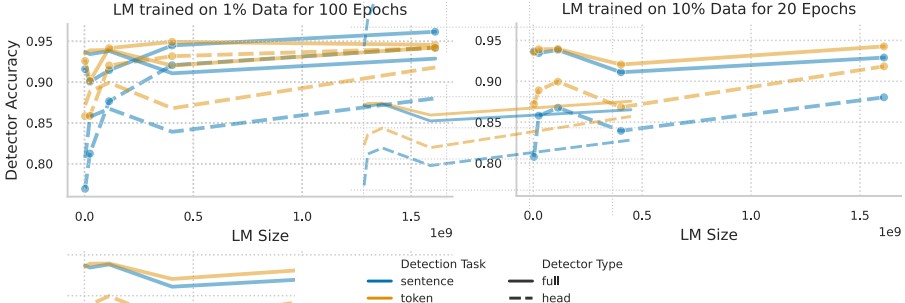

Figure 5: **Hallucination detection accuracy as a function of the LM size** for various task formulations and detector types. Detectors were trained and evaluated on distinct splits of data obtained by having a given pretrained LM generate 5 completions for every subject-predicate in its training set (using temp = 1.0). The accuracy of all the trained hallucination detectors is generally high, especially for outputs of the larger LMs. Larger (full) detectors work better than smaller ones (head). The token-level detection task formulation seems to provide higher detection accuracy, although not in all cases. The results here are confounded by the varying hallucination rates of the underlying LM (e.g., if the LM hallucinates only 5% of the time, a detector which finds no hallucinations achieves 95% accuracy).

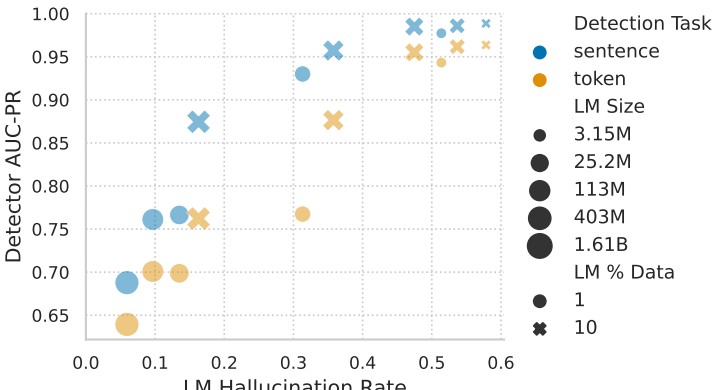

Figure 6: **AUC-PR as function of LM hallucination rate** for the full detectors. Same setup as in Figure 5, except LM size is now represented by the marker size. Showing results for data generated by LMs trained for 100 (resp. 20) epochs on 1% (resp. 10%) of the data. AUC-PR does not depend on the proportion of hallucinations in the evaluation data (i.e., the LM's hallucination rate), thus providing providing a better measure of the detector's ability to catch hallucinations. Unlike for accuracy (Figure 5), the sentence task is clearly superior in AUC-PR terms (can also be seen in Figure 9), although better token performance can be attained by attaching the detector to a different LM layer (Figure 10). More importantly, the detectability of hallucinations is *inversely* proportional to the LM size (largest dots/LMs in bottom left, smallest in top right). Larger LMs have lower hallucination rates, but it's also the harder to detect their hallucinations. This can be seen even more clearly in Figure 7.

## 5 Limitations

Several factors may limit correspondence of our results to behaviour of state-of-the-art (SOTA) LMs. Firstly, the KG differs from the data normally used for LM training (e.g., no repetition of facts in the corpora, no semantic ambiguity, simple sentence structure). Secondly, the LMs we train are significantly smaller than SOTA LMs, for which we adjust by using a proportionally smaller dataset. While we believe the qualitative interpretation of our results would generalize, we cannot rule out appearance of emergent capabilities (Wei et al., 2022) which may, e.g., significantly reduce the hallucination rate on unseen data. Thirdly, we only study hallucinations that result in not remembering something that

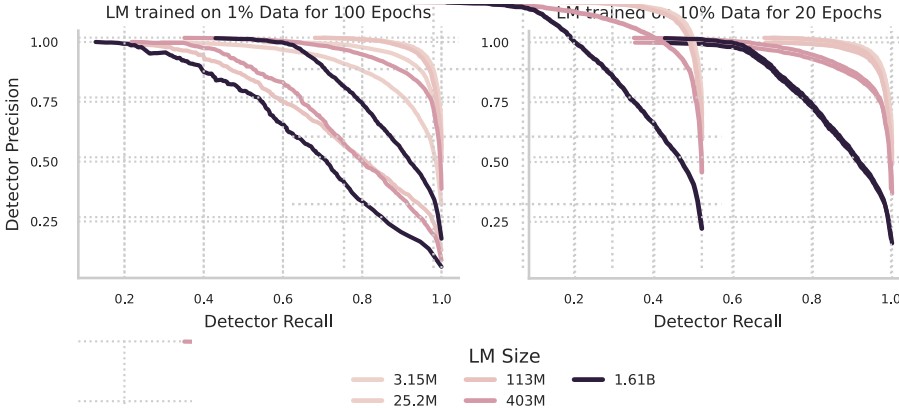

Figure 7: **Precision-Recall curves for the full detectors** trained on the sentence task. These curves correspond to the relevant subset of the AUC-PR values from Figure 6. The larger the LM, the harder it is to detect its hallucinations.

appeared verbatim in the training data. While this has enabled most of our analysis, some of our conclusions may not translate to other types of hallucinations.

For hallucination detectors, we focused on general trends rather than exhaustive coverage of existing methods, and it is possible that alternative methods could have yielded better results. However, the qualitative consistency of the results relating to dependence of detector performance on LM scale suggests these are general properties of hallucination detectors. Finally, dataset imbalance driven by varying hallucination rates between our LMs had a strong effect on performance of the hallucination detectors. We did not experiment with methods to adjust for such imbalance, beyond implicitly measuring performance at different score cut-offs via AUC-PR and the PR curves.

## 6 Conclusion

We explored the relationship between LM hallucinations and their scale. In particular, our paper sheds light on what it would take to eliminate factual hallucinations in the case where the correct answers—and no wrong answers—appear verbatim in the training set. Our study can also be seen as studying how much computation is needed for an LM to memorize all the knowledge content of a dataset. For a fixed dataset size, larger longer trained LMs hallucinate less. However, we found increasing dataset size also increased hallucination rate, given a fixed number of training epochs and LM size. This is because a larger dataset means more facts to learn. Similar to Kandpal et al. (2023), we hypothesise that standard LM training data contains repeated facts—and other repetitive text structures that do not require hard memorization—which makes the effect of dataset size on loss positive.

We found achieving a ≤5% hallucination rate on training data requires LMs much larger than currently considered optimal (Hoffmann et al., 2022), trained for much longer (20+ epochs). Moreover, the longer training can hurt LM generalisation, presenting a trade-off between hallucination rate and other performance metrics.

Given the difficulty of eliminating hallucinations outright, we explored how detectability of LM hallucinations depends on scale. We found an inverse relationship between LM scale and the detectability of its hallucinations. That is, while larger, longer-trained LMs hallucinate less, detecting their remaining hallucinations becomes increasingly hard.

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

# Appendix

## A  Related Work

Hallucination in the context of natural language processing (NLP) is usually defined as the phenomenon of model-generated text that is seemingly fluent and coherent, but is actually nonsensical or unfaithful to the provided source content (Ji et al., 2023). Although the study of hallucinations in LMs is nascent, there is significant interest due to the challenges and risks they pose. Here we discuss a variety of relevant work, but direct the reader to (Ji et al., 2023) for a more comprehensive survey.

Kandpal et al. (2023) found that LMs struggle to learn long-tail knowledge that is not seen by the model multiple times during training. This is consistent with our finding that multiple passes over triplets are needed for the model to memorize facts. Kadavath et al. (2022) performed a large scale study to determine if models can give an indication of whether they will give a correct response to a multiple choice question. They found that for the majority of questions, the model was well calibrated in predicting correctness, with the exception of long-tail or out-of-distribution data.

Carlini et al. (2023) study memorization in LMs and, similarly to this work, find that models memorize more with respect to scale and the number of times a concept is seen during training. We find that even in larger models, the hallucination rate remains above 5% on the training set even after 100 passes over the data. Dziri et al. (2022) found that the phenomenon of hallucinations is compounded by datasets and benchmarks which are themselves rife with hallucinated data. Agrawal et al. (2023) study hallucinations of references and find that inconsistency of multiple sampled references is a strong indicator of hallucinations.

Some recent work (Guu et al., 2023; Grosse et al., 2023) has developed methods to attribute predictions to individual training examples using influence functions. However, computing influence functions exhaustively over the training data remains prohibitively expensive. Therefore, we study hallucinations in the setting where we can know and control relationships between facts contained in the data that is exposed to the model.

Mitigation techniques range from automated model rewrites (Dhuliawala et al., 2023; OpenAI, 2023), attribution and grounding (Rashkin et al., 2021), to classification (probing) based on model internals. Dziri et al. (2021) use a critic and retrieval augmented dialogue model to retrieve facts from a KG. Li et al. (2023) use linear probes to detect internal model attention heads that are more likely to indicate truthfulness, and use this to push the model more towards truthful responses. In this work, we try to assess, in a best case scenario, how well either works to mitigate hallucinations.

## B  Learning rates, number of training steps

As mentioned in Section 2.2, the base learning rate for each model size is set to be a constant (a hyperparameter searched over 2.5, 5, and 10) divided by the square root of the number of LM's non-embedding parameters. The resulting exact learning rates are listed in Table 3.

| Model size | Learning rate |
|---|---|
| 3.15M | 0.002885 |
| 25.2M | 0.001 |
| 113M | 0.00047 |
| 403M | 0.00025 |
| 1.61B | 0.00011 |

Table 3: Base learning rates used for various model sizes.

| Data subset | 2.6K | 5.2K | 26K | 52K | 260K | 520K |
|---|---|---|---|---|---|---|
| FVS + PVS, 1% | ✓ | ✓ | ✓ | ✓ | ✓ | ✓ |
| FVS + PVS, 10% | ✗ | ✗ | ✓ | ✓ | ✓ | ✓ |
| FVS + PVS, 100% | ✗ | ✗ | ✗ | ✗ | ✓ | ✓ |

Table 4: Number of training steps for different data sizes.

## C  Additional plots

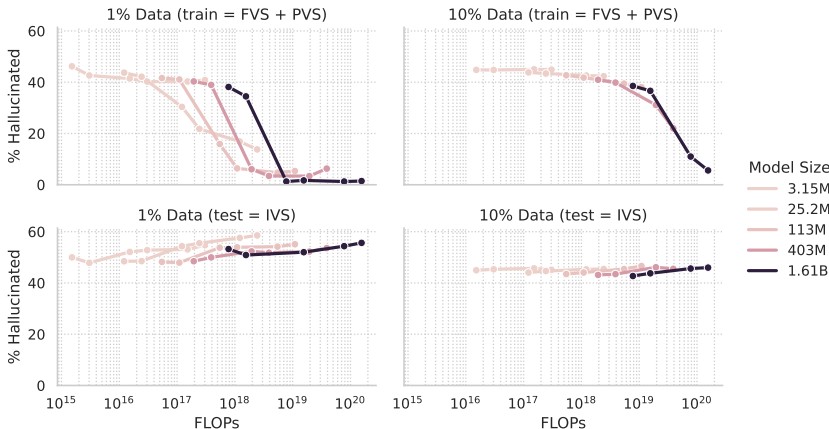

Figure 8: **Hallucination rate per LM training FLOPs** on examples seen (top) and not seen (bottom) during training. Same as Figure 2, except for using temp = 0 instead of temp = 1 to generate the samples. Note the more pronounced decay in out-of-distribution (IVS) performance with length of training, emphasising the possible trade-off between training set hallucination rate and

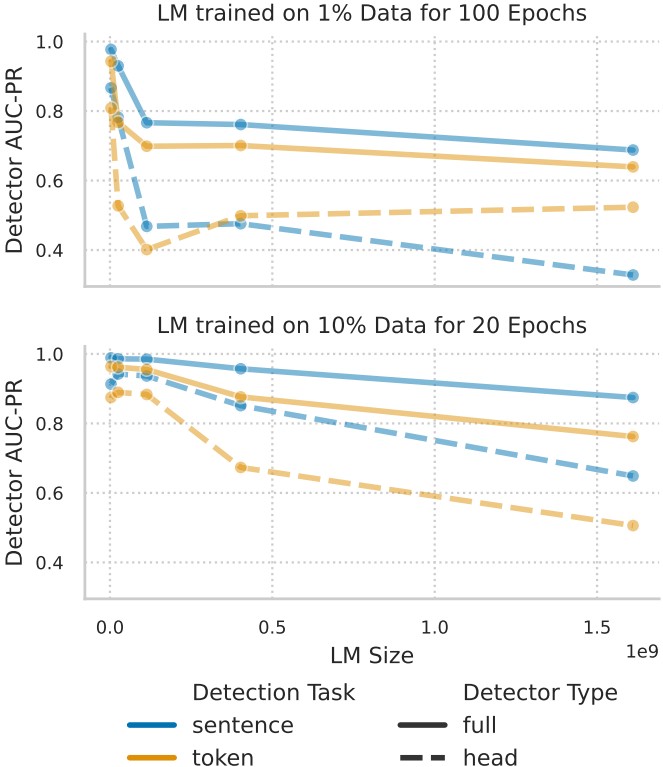

Figure 9: **AUC-PR as function of LM size** for various task formulations and detector architectures. Same setup as in Figure 5. Complementing Figure 6, shows the sentence task is superior in terms of AUC-PR. However, the token task performance could be improved by attaching the detector to a different layer of the LM (Figure 10).

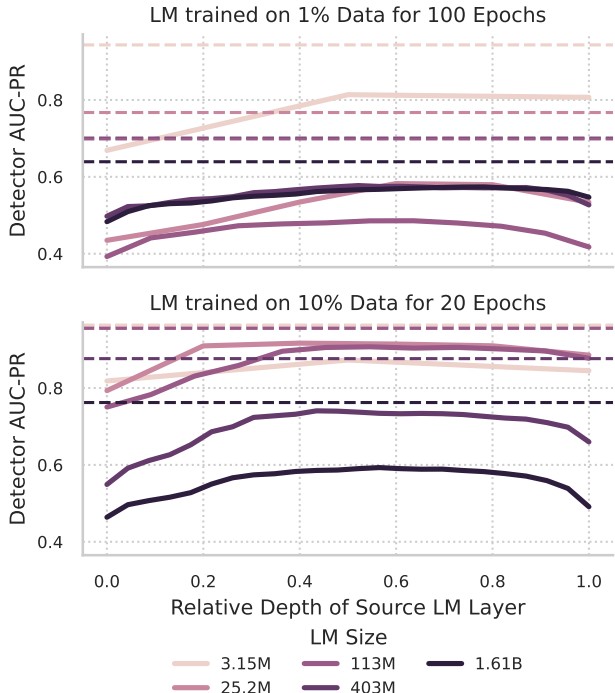

Figure 10: **AUC-PR of head detectors in the token setup as function of LM layer depth.** We fitted a distinct head detector using token representations from different layers of the LM, specifically from after each transformer block. The AUC-PR is the lowest near LM input, then grows, before often dipping again near LM output. This is interesting given that head detectors are typically used with top layer embeddings (e.g., Kadavath et al., 2022; Li et al., 2023). We thus only report the top layer embedding performance of head detectors throughout Section 4.2, but the reader should remember these results are typically few percent lower than the best possible.

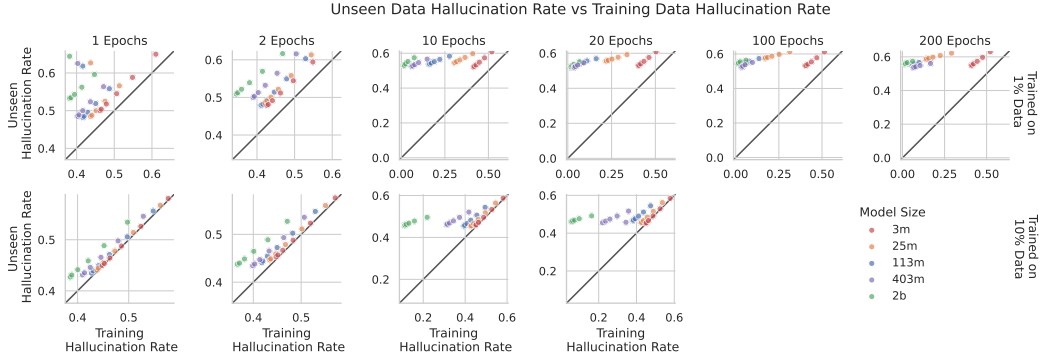

Figure 11: **Hallucination rate for held-out unseen data vs. the hallucination rate for training data.** Models always hallucinate more on data it has not seen before. This is especially true after several training epochs, when the LMs presumably start to memorize more.

