# OpenReview forum: "Training Language Models on the Knowledge Graph: Insights on Hallucinations and Their Detectability"
_colmweb.org/COLM/2024/Conference — COLM_

### Official Review · Reviewer_cjCt · 2024-04-12

**Rating:** 6
**Confidence:** 4
**Ethics Flag:** 1

**Summary:**

This paper presents a comprehensive study on the relationship between LM hallucinations and their scale. The authors construct a knowledge graph (KG) based dataset to train LMs, creating a controllable experimental environment to study hallucinations. The authors also design several hallucination detectors to study how hallucination detectors depend on scale (e.g., LM size, training epochs, datasets).

**Reasons To Accept:**

The presentation of the methods and experiments is clear and methodical. And this paper provides a comprehensive understanding of the hallucination problem with the scale. The results and findings presented in the paper are insightful and interesting, for example, this paper reveals a trade-off between the model's ability to generalize and its tendency to hallucinate.

**Reasons To Reject:**

1.It will be better if the authors could provide more detail analysis about the causes of hallucinations, such as noise in the dataset (training on KG may limit this setting), etc., may provide richer insights. I also wonder what will happens if training LMs on KG with incorporating textual context.

2.Providing some case studies of hallucination examples on the scale of LMs can help readers to understand the nature of hallucination and the working mechanism of the detector more intuitively.

3.The Related Work section should be included in the main tex, rather than in the Appendix. And this paper mentions some correlations with existing methods but does not provide direct comparisons.

---

> ### Author Rebuttal · Authors · 2024-05-29
>
> We thank the reviewer for calling our findings insightful and interesting, and appreciating the comprehensiveness of our investigation. We agree that our work has limitations, but we believe that it still makes a significant and novel contribution to the field of LM research.
>
> We hope to address your reasons for rejection below.
>
> ## Reason 1
> We experimented with creating a taxonomy of hallucination causes during our experiments, but found the categories did not transfer well between model sizes and datasets (likely because of the rather larger performance differences). We are not aware of noise issues in the KG, but it would be an exciting direction of future research to try to procedurally generate triplets/sentences with incorrect objects, and measure the impact of such contamination on model performance. Another direction would be to contaminate the hallucination labels the detectors are trained on, and measure its impact. Our work trains LMs on KG entirely, but we can imagine a future where KG is combined with linguistic data; for more discussions of generalizability of our conclusions to natural language, please see our reply to Reviewer o4Hv.
>
> ## Reason 2
> Thanks for a great suggestion! We agree that providing some case studies of hallucination examples will really help drive our points home. Here are two examples, note that we removed the identifying information for simplicity.
>
> Input: '<S_TKN> A specific unnamed language <P_TKN> Main places spoken <O_TKN>'
> Ground truth answer: ['United States Regions']
> Output by model: ['Italy', 'Papua New Guinea']
>
> Input: '<S_TKN> A specific album name <P_TKN> Genre <O_TKN>'
> Ground truth answer: ['Indie rock', 'Indie pop']
> Output by model: ['Indie game']
>
> As you can see that while the model does hallucinate (make up wrong answers), the output is still semantically reasonably adjacent to the ground truth answer.
>
> ## Reason 3
> Thank you for your valuable suggestion. We agree that Related Work should be in the main text. If accepted, we will use the additional page to include related work in the camera ready version. Comparing to existing methods (e.g. methods for mitigating hallucinations) across model and training settings will be challenging, but it is a valuable direction worth thinking about.
>
> Thank you for your constructive suggestions, and please let us know if we can answer any more questions.

---

> > ### Comment · Reviewer_cjCt · 2024-06-03
> > **Thanks for your response.**
> >
> > Thanks for your response. The replies for Reason 1 and 3 have partially solved my concerns, and I would like to keep my score for the current version. In addition, by "hallucination examples on the scale of LMs" I mean the examples by LMs on different scales (e.g., LM size, training epochs, datasets).

---

> > > ### Author Response · Authors · 2024-06-05
> > >
> > > That's a fantastic suggestion! We will include the same example and their predictions by LMs with various model sizes, training data sizes, and training lengths in the revised draft. Thank you!

---

### Official Review · Reviewer_MRVp · 2024-05-08

**Rating:** 6
**Confidence:** 4
**Ethics Flag:** 1

**Summary:**

The paper examines hallucinations in language models trained on knowledge graphs, focusing on cases where correct answers are present in the training data. It reveals that larger models have lower hallucination rates but require more computational power. The study also finds that the detectability of hallucinations decreases as model size increases, suggesting a trade-off between model scale and generalization capabilities.

**Questions To Authors:**

1. Could the authors please elaborate on the necessity of incorporating special tokens during the training phase, and discuss whether the deviation from the natural language text format could potentially influence the conclusions drawn from the study?

2. The results presented in some of the figures, such as Figures 4 and 7, are not as intuitive as they could be, making it challenging to derive clear conclusions. With the presence of multiple variables in these charts, it seems difficult to reach a valid conclusion. The authors should consider revising the presentation to allow for a more straightforward interpretation of the results.

3. In Appendix B, I notice that the authors used different learning rates for models of different sizes. But apparently, the values of these learning rates are very strange, how did the authors get these particular learning rate values (e.g., 0.00047 for the 113M model)?

**Reasons To Accept:**

1. The approach of training LMs from scratch using knowledge graphs is innovative and brings a fresh perspective to the study.
2. The paper presents intriguing conclusions that hold significant implications for future research in the field.

**Reasons To Reject:**

1. This paper studies the phenomenon of hallucinations, particularly focusing on the task of predicting the object entity in a triplet. However, the research scenario presented in this work is relatively singular, primarily concentrating on the direct prediction of facts, which seems limited compared to the broader types of hallucinations observed in real-world tasks, such as text generation, dialogue systems, summarization, translation, etc. As a result, the conclusions drawn from this study may lack universality and may not adequately reflect the performance of language models across a diverse array of tasks.

2. Some findings in this paper are interesting but potentially insufficient and possibly incorrect. For example, the analysis presented in Figures 2 and 3 indicates that an increase in dataset size leads to a higher rate of hallucinations. However, this might be due to the models not being sufficiently converged when the same amounts of FLOPs are applied across different dataset sizes. To achieve more reliable conclusions, models should be allowed more FLOPs for larger datasets to ensure adequate training and convergence.

3. The manuscript's complexity and lack of clarity in certain experimental descriptions make it difficult to follow. The figures, with their numerous variables, are not clearly articulated, hindering the extraction of useful conclusions. The paper would benefit from a more concise presentation and clearer figure design to enhance readability and comprehension.

---

> ### Author Rebuttal · Authors · 2024-05-29
>
> We thank the reviewer for acknowledging the novelty and significance of our work. We agree that our work has limitations, but we believe that it still makes a significant and novel contribution. First, we answer your questions:
>
> ## Q1
> The first and last tokens are just like BOS and EOS. P_TKN and O_TKN (Fig 1) delineate individual triplet parts, preventing ambiguities that could confound evaluation (e.g. in case of prefix overlap). These tokens may simplify the task, similar to eg, arXiv:2301.13195, arXiv:2310.02226, arXiv:2309.16588, but we think the task is still difficult given the need to train larger than Chinchilla optimal models for many epochs. For a discussion of generalizability of our conclusions to natural language, please see our reply to Reviewer o4Hv and tRfu (Reason 2).
>
> ## Q2
> In Fig 4, there are many variables in play: LM size (marker size), training length (dot vs cross), temperature (color), but the overall trend is consistent: the longer the training, the better the PR. The larger the LM size, the better the PR. Lower temperature -> higher precision. Higher temperature -> higher recall. We will make those observations clearer.
>
> ## Q3
> Learning rates are lowered as model size increases for training stability. We use this formula stated in Section 3.2: learning rate = constant / sqrt(number of non-embedding parameters).
>
> Next, we hope to address your reasons for rejection.
>
> ## Reason 1
> You’re right that we focus only on one hallucination type at the moment, in a controlled environment with special data. You can consider this a “lab” setting to gain insights and understanding. We think that thoroughly understanding even a single type of hallucinations in a simplified setting is a crucial first step to understanding many other types of hallucinations in other real-world settings.
>
> ## Reason 2
> You’re correct, we will highlight more data requires more compute! The data and model sizes we used are due to the rather challenging nature of the KG dataset (see our reply to Reason 3 of Reviewer tRfu). In short, achieving non-trivial hallucination rate requires an order of magnitude bigger models trained for much longer than Chinchilla optimal. With limited resources, we studied how different ways of "overscaling" affect hallucination rates, illuminating the scale of the challenge.
>
> ## Reason 3
> Thanks, we will try to improve our presentation. For example, see answer to Q2 above.

---

> > ### Comment · Reviewer_MRVp · 2024-06-05
> >
> > Thank you for your detailed response, which addresses most of my concerns. However, the paper should focus on making its conclusions accurate and faithful. I will keep my positive rating and hope that the revised paper is better.

---

### Official Review · Reviewer_tRfu · 2024-05-10

**Rating:** 7
**Confidence:** 4
**Ethics Flag:** 1

**Summary:**

This paper investigates the scaling law of language models (LMs) and hallucination. The authors control the train/test facts by constructing a knowledge graph (KG) based dataset and train language models from scratch. They ablate LMs of varying sizes and evaluate their hallucination rates on seen and unseen data. They find that while increasing the LM size and training compute reduces hallucinations on the training set, eliminating hallucinations entirely requires an order of magnitude more compute than currently considered optimal. They also study hallucination detectors and find an inverse relationship between LM scale and the detectability of hallucinations by these detectors.

**Questions To Authors:**

1. Can the authors comment on how the findings may change when considering more advanced language understanding capabilities needed for hallucinations in natural language?
2. What are some ways the KG-based approach could be extended to study other types of hallucinations beyond factual knowledge verbatim in the training set?

**Reasons To Accept:**

1. The use of a controlled KG-based dataset is a novel approach that enables quantifying hallucinations in a precise manner not applicable to previous data. This controlled setup reveals interesting insights.
2. Comprehensive analysis of how hallucination rates depend on LM scale, training compute, and dataset size.
3. Investigation into the scaling properties of hallucination detectors and their ability to catch hallucinations from LMs of different scales. The inverse scaling of hallucination detectability with LM scale is an interesting result with potential ramifications for detector-based mitigation strategies.
4. Clear writing and good visualization of key results.

**Reasons To Reject:**

1. Limited to studying hallucinations on factual knowledge present verbatim in the training data (which is not natural language sentences). Other types of hallucinations are not explored.
2. Some discussion on similarities/differences compared to natural language could have strengthened the connection to real-world LM applications.
3. LM and dataset sizes studied are still significantly smaller than state-of-the-art LLMs, raising potential generalizability concerns.

---

> ### Author Rebuttal · Authors · 2024-05-29
>
> We thank the reviewer for pointing out our comprehensive analysis of scaling of LMs and hallucination detectors, the clear writing and visualization, and that our setup enables quantifying hallucinations in a previously impossible precise manner.
>
> First, we answer your questions:
>
> ## Q1
> This is an intriguing question. Could the reviewer please clarify what they mean by “advanced language understanding capabilities” in this case? Perhaps we can better answer this question given an example of such capability.
>
> ## Q2
> We appreciate this thought-provoking question! Logical/Counterfactual hallucinations could also be studied if we extend the KG-based approach by introducing conflicting <subject>,<predicate> pairs during inference. One could also add contamination to the dataset with generated untruths to investigate their impact on hallucinations. One could also add contradictions and various kinds of structured noise one might find in natural language online.
>
> Next, we hope to address your reasons for rejection.
>
> ## Reason 1
> This study focuses on hallucination instances where a model is violating the facts it is trained on, because it is easy to measure, and because understanding this type of hallucination is a crucial first step to understanding many other types of hallucinations. We are open to extending other types of hallucinations as mentioned above in answer to Q2.
>
> ## Reason 2
> Training LMs on KG is fundamentally not that different from training on language; both aim for next-token prediction and generalization to OOD data. Our setting just simplifies it: there is a  clear definition of the two distributions, and complexities arising from linguistic data are removed by using structured data.
>
> ## Reason 3
> We found that the LMs we trained hallucinate a lot even on these small datasets, even when trained for much longer than usual. Further, if we take the ~20 params/token Chinchilla optimal ratio, the 1.6B model trained on 10% of the data would have been roughly optimal (see Table 1), but gets ~50% hallucination rate on the data seen in training (Figure 2, top right), which is too high to be the only object of our investigation. Extrapolating the pattern from Figure 2, we realized that to train a proportionately larger LM on a larger dataset to the point where it doesn't hallucinate a lot is going to be too costly, and such compute budget could be better spent elsewhere.
>
> Please let us know if you have any further questions or comments.

---

> > ### Comment · Reviewer_tRfu · 2024-06-03
> >
> > Thanks for the response,  the thoughts on extending the KG-based approach look interesting!
> >
> > By "advanced language understanding capabilities" I mean the ability for reasoning like doing math, physics, writing code, and long context like writing the full novel.

---

> > > ### Author Response · Authors · 2024-06-05
> > >
> > > Thanks for clarifying the notion of "advanced language understanding capabilities." To that end, we believe our findings based on KG will still hold: more capable LMs will hallucinate less, but the hallucinations produced by them will be harder to detect. Moreover, to eliminate hallucination completely will be difficult, possibly requiring orders of magnitudes more FLOPs than what's currently considered optimal.

---

### Official Review · Reviewer_o4Hv · 2024-05-11

**Rating:** 7
**Confidence:** 2
**Ethics Flag:** 1

**Summary:**

The paper explores the relationship between LM hallucinations and their scale, focusing on eliminating hallucinations where correct answers are in the training set. To this end, the authors train a set of LMs on a KG-based dataset. The major conclusions include: 1) Larger and longer trained LMs hallucinate less on a fixed dataset, but increasing dataset size can increase hallucination rate. 2) Achieving a low hallucination rate requires LMs much larger than currently considered optimal trained for longer periods, with a trade-off between hallucination rate and other performance metrics. 3) Detectability of LM hallucinations decreases as LM size increases.

**Reasons To Accept:**

* The paper is generally well-organized and clearly written.
* The paper explores an important topic with reasonable experiments and analysis.
* The conclusions of the paper provide potentially useful knowledge regarding developing practical solutions to reduce LM hallucinations.

**Reasons To Reject:**

* My major concern is that the data used for training LMs in this paper is quite different from the data used for LM training in practice. While the paper has a short discussion about it in the limitations section, it would make the paper stronger if the authors could provide empirical evidence to justify the conclusions' generalizability.

---

> ### Author Rebuttal · Authors · 2024-05-29
>
> We thank the reviewer for acknowledging that our manuscript addresses an important topic, is well written, and could provide insights for practical solutions.
>
> We further appreciate the reviewer's concerns about the generalizability of our conclusions, and agree that the data used for training LMs in this paper is different from the usual natural language data. Our training data is entirely extracted from a KG, a structured representation of factual knowledge, and formatted to suit LM training. We purposefully adopted this setting, because it allows for a fully controlled data environment where we can control what knowledge the LM was exposed to during training, and thus make a clear separation of knowledge pieces between training and test. This hard separation is difficult, to the point of impossible, to ensure with natural language data, where the “unit” of knowledge does not have a clear definition. To exclude a piece of knowledge entirely from training means that every possible linguistic variation of such knowledge needs to be excluded. It thus poses a great challenge to fully control the factual content in natural language data. We therefore resort to KG-based LM training to seek a deeper understanding of the hallucination effect.
>
> Furthermore, training LMs on KG is fundamentally not that different from training on language. In the end, we are training LMs to perform next-token-prediction and study its generalizability, on in-distribution and out-of-distribution data. Our setting simplifies the process by having a clear definition of the two distributions, removes the complexities that come with linguistic data, and comes up with a simple format to handle structured data.
>
> At the moment, training on KG triplets may seem to go against the current standard of LM training, however, the field of LM training is evolving fast, and since KGs are already widely used in a variety of applications alongside natural-language LMs, including question answering, information retrieval, and knowledge discovery, it may not be long before we witness alternative ways of training LMs employing a wide range of data modalities. In one of the many possible futures of LM training, this setting will not seem so odd.
>
> Therefore, we believe that our conclusions are generalizable to current and future LMs. Please let us know if you have any more questions or comments.

---

> > ### Comment · Reviewer_o4Hv · 2024-06-05
> >
> > Thanks for the response. I still suggest having empirical evidence to justify the conclusions' generalizability and would like to keep my score.

---

### Decision · Program_Chairs · 2024-07-10

**Decision:**

Accept

**Comment:**

Metareview:
The paper investigates the relationship between LM hallucinations and their scale, focusing on eliminating hallucinations where correct answers are in the training set. The authors train a set of LMs on a KG-based dataset, concluding that larger and longer-trained LMs hallucinate less on a fixed dataset, but increasing dataset size can increase hallucination rates.
Reasons To Accept:
1. The paper is well-organized and clearly written.
2. It explores an important topic with reasonable experiments and analysis.
3. The conclusions provide potentially useful insights for developing practical solutions to reduce LM hallucinations.
Reasons To Reject:
1. The data used for training LMs in this paper is different from the data used for LM training in practice, raising concerns about the generalizability of the conclusions.
2. The research scenario is relatively singular, focusing on the direct prediction of facts, which seems limited compared to the broader types of hallucinations observed in real-world tasks.
3. The analysis indicates that increasing dataset size leads to higher hallucination rates, but this might be due to models not being sufficiently converged.